# Vitamin B_12_ Ameliorates Pesticide-Induced Sociability Impairment in Zebrafish (*Danio rerio*): A Prospective Controlled Intervention Study

**DOI:** 10.3390/ani14030405

**Published:** 2024-01-26

**Authors:** Madalina Andreea Robea, Ovidiu Dumitru Ilie, Mircea Nicusor Nicoara, Gheorghe Solcan, Laura Ecaterina Romila, Dorel Ureche, Alin Ciobica

**Affiliations:** 1Doctoral School of Biology, Faculty of Biology, “Alexandru Ioan Cuza” University of Iasi, Bd. Carol I, 20A, 700505 Iasi, Romania; madalina.robea11@gmail.com; 2Faculty of Medicine, University of Medicine and Pharmacy “Grigore T. Popa”, University Street No. 16, 700115 Iasi, Romania; ovidiuilie90@yahoo.com; 3Department of Biology, Faculty of Biology, “Alexandru Ioan Cuza” University of Iasi, Bd. Carol I, 20A, 700505 Iasi, Romania; alin.ciobica@uaic.ro; 4Doctoral School of Geosciences, Faculty of Geography and Geology, “Alexandru Ioan Cuza” University of Iasi, 700505 Iasi, Romania; 5Internal Medicine Clinic, Faculty of Veterinary Medicine, Ion Ionescu de la Brad Iasi University of Life Sciences, 700489 Iasi, Romania; gsolcan@yahoo.com; 6Preclinical Department, Apollonia University, Pacurari Street 11, 700511 Iasi, Romania; laura_ekaa@yahoo.com; 7Department of Biology, Ecology and Environmental Protection, Faculty of Sciences, University “Vasile Alecsandri“ of Bacau, Calea Marasesti Street, No. 157, 600115 Bacau, Romania; 8Academy of Romanian Scientists, 54, Independence Street, Sector 5, 050094 Bucharest, Romania; 9Center of Biomedical Research, Romanian Academy, Iasi Branch, Teodor Codrescu 2, 700481 Iasi, Romania

**Keywords:** *Danio rerio*, behavior, autism spectrum disorder, pesticide, vitamin B_12_, oxidative stress

## Abstract

**Simple Summary:**

Dietary vitamin supplementation is frequently mentioned as an alternative therapy for people with autism but is still controversial due to a lack of studies. The current study described a hypothetical real-life situation and evaluated the effect of vitamin B_12_ on the behavior of zebrafish. Vitamin-treated fish showed behavioral improvements after 14 days of daily exposure. In addition, the presence of the vitamin improved the redox state, leading to increased activity of antioxidant enzymes.

**Abstract:**

Constant exposure to a variety of environmental factors has become increasingly problematic. A variety of illnesses are initiated or aided by the presence of certain perturbing factors. In the case of autism spectrum disorder, the environmental component plays an important part in determining the overall picture. Moreover, the lack of therapies to relieve existing symptoms complicates the fight against this condition. As a result, animal models have been used to make biomedical research easier and more suited for disease investigations. The current study used zebrafish as an animal model to mimic a real-life scenario: acute exposure to an increased dose of pesticides, followed by prospective intervention-based therapy with vitamin B_12_ (vit. B_12_). It is known that vit. B_12_ is involved in brain function nerve tissue, and red blood cell formation. Aside from this, the role of vit. B_12_ in the redox processes is recognized for its help against free radicals. To investigate the effect of vit. B_12_, fish were divided into four different groups and exposed to a pesticide mixture (600 μg L^−1^ fipronil + 600 μg L^−1^ pyriproxyfen) and 0.24 μg L^−1^ vit. B_12_ for 14 days. The impact of the compounds was assessed daily with EthoVision XT 11.5 software for behavioral observations, especially for sociability, quantified by the social interaction test. In addition, at the end of the study, the activities of superoxide dismutase (SOD), glutathione peroxidase (GPx), and malondialdehyde (MDA) were measured. The results showed significant improvements in locomotor activity parameters and a positive influence of the vitamin on sociability. Regarding the state of oxidative stress, high activity was found for SOD and GPx in the case of vit. B_12_, while fish exposed to the mixture of pesticides and vit. B_12_ had a lower level of MDA. In conclusion, the study provides new data about the effect of vit. B_12_ in zebrafish, highlighting the potential use of vitamin supplementation to maintain and support the function of the organism.

## 1. Introduction

Being one of the eight vitamins of group B, vitamin B_12_ (vit. B_12_), also called cobalamin due to the presence of the mineral cobalt in its structure, has often been mentioned as essential for cellular metabolism [1]. It is known to be involved in the synthesis of DNA molecules and myelin but also intervenes in the mediation of oxidative stress [2]. Its deficiency in the body has been linked to motor alterations, memory loss, irritability, poor balance, and cognitive impairment [1,2,3]. It has frequently been associated with increased anemia; alterations in the central or peripheral nervous system; and the onset of mechanisms prior to disorders such as autism spectrum disorder (ASD), schizophrenia, and epilepsy [3,4,5]. Despite the established recommended intake, which can vary across countries and continents, vit. B_12_ levels can be recorded at different values that indicate a deficiency. For instance, children, adolescents, women of childbearing age, and the elderly are considered vulnerable to vit. B_12_ deficiency, according to Vargas-Uricoechea et al. [6]. With an underrecognized capacity to be considered a serious disorder, this deficiency can be heightened by autoimmune conditions, alcohol consumption, malabsorption, or a dietary insufficiency (e.g., in vegetarians) [7]. Additionally, vit. B_12_ serves as a cofactor for two enzymes [8]. Methionine synthase in the cytoplasm requires vit. B_12_ in the form of methylcobalamin to catalyze the conversion of homocysteine to methionine. If the process is impaired due to a lack of vit. B_12_, the intracellular homocysteine level increases, homocysteine being implicated in the mediation of ROS buildup, e.g., homocysteine auto-oxidation [9]. Vit. B_12_ is also essential for the mitochondrial enzyme methylmalonyl CoA mutase, which converts methylmalonyl CoA to succinyl CoA, a step in the oxidation of odd-chain fatty acids and the catabolism of ketogenic amino acids [1].

Mentioned for the first time in 1985, the concept of ”oxidative stress” has been defined as an imbalance between oxidants and antioxidants, characterizing the capacity of an organism to maintain its physiological state [10,11,12]. Usually, increased levels of reactive oxygen species (ROS) are linked to important changes in cell functioning that lead to biomolecule damage and, in the end, can be a determining factor causing cell apoptosis [13,14]. It has been indicated that there is a strong correlation between ASD and oxidative stress [15].

Recognized through the presence of significant disruptions in speech and social skills and through repetitive behavior, ASD is a well-known neurodevelopmental illness that started to gain more attention due to its high incidence [16,17]. Recent data have highlighted the prevalence of ASD among US children, estimated at 1 in 36 [18]. The rate of prevalence depends on several variables. For instance, it has been shown that the frequency of ASD is greater for Asian children, non-Hispanic Black children, children with higher Social Vulnerability Index scores, and children who receive treatment in urban primary care locations [19]. Additionally, the sex-ratio difference is another concern among specialists. There are reports that present a much higher incidence for boys than girls in detecting ASD, and this is due to the complicated process needed to discover the specific features of autism in girls [20]. In real-world practice, according to the Diagnostic and Statistical Manual of Mental Disorders (DSM-V), ASD is diagnosed based on three main characteristics: difficulty in social communication; deficiencies in social relationships; and specific, repetitive patterns of behavior, activities or interests [21,22,23]. Although its etiology is not fully understood, ASD may be seen from a complex perspective that combines genetic and environmental components [17,24,25,26]. The most common cause is genetic susceptibility, although prenatal stress, infections, parental age, dysfunctional familial relationships, and exposure to neurotoxic chemicals are all believed to be risk factors [27,28,29,30,31].

For instance, the lack of a sufficient amount of cobalamin in the bodies of autistic people is mainly due to poor nutrition, specifically in terms of micronutrients, but this is not the only cause [4,32]. Data on the prevalence of clinical eating issues such as anorexia nervosa and bulimia nervosa in the population with autism are limited, but the latest results indicate that these conditions are more common in people with autism and/or attention-deficit/hyperactivity disorder (ADHD) than in the general population [33,34,35,36]. Moreover, screening for vit. B_12_ in children diagnosed with ASD showed lower vitamin levels in those children than in the control group, according to several studies [37,38,39,40]. Methylcobalamin supplementation has been proven to be effective for participants in several studies carried out, leading to a relief of autism-related symptoms [41,42,43,44,45]. Even though little research has been conducted on the effect of vit. B_12_ in ASD or its effectiveness in alleviating core or associated symptoms, it appears that the current evidence supports the vitamin’s capacity to help and even to be regarded as a prospective treatment intervention [45].

The role of vit. B_12_ has long been recognized, starting with the preliminary data obtained from experimental research, whose outcomes contributed to new data acquisition. For instance, vitamin activity has been investigated in several studies that used animals as model organisms (zebrafish, mice, rats) for a specific disorder. Although an animal model cannot mimic 100% of human features, there are three rules for validating its suitability as a model; it should have similar causes or mechanisms of onset, similar symptoms, and similar responsiveness to treatment [46,47]. In this study, the zebrafish (*Danio rerio*) was chosen due to its multiple advantages, and it was considered to be eligible for development as an animal model for ASD, a fact highlighted by numerous reports [48,49,50,51,52]. In addition, it possesses the full complement of cobalamin metabolic enzymes, and a deficiency of this nutrient can be caused by mutations in mmachc, a gene responsible for vit. B_12_ activity regulation [8,53,54]. According to Sloan et al. [53], zebrafish with mmachc mutant alleles showed impaired growth and developmental delays but also responded to established therapies, suggesting that this may be a suitable model for studying cobalamin deficiency. A 2012 study evaluated the effect of various vit. B_12_ levels in zebrafish and demonstrated that 5 µg vit. B_12_ kg^−1^ was insufficient to support whole-body vitamin storage, but there were also no signs of deficiency [55]. When vit. B_12_ was administered in a mixture with 100 µg L^−1^ malathion, an organophosphate insecticide, it led to an arousal of the motor functions by modulating acetylcholinesterase activity. These results were further supported by the diminished oxidative-stress status [56].

Aside from the genetic component, the environment has recently attracted increased attention as a risk factor for ASD. Specialists started to point out the involvement of the environment in ASD and the negative impact [16,24,28,30]. An imbalance between excitatory and inhibitory neuronal activity in most cases of ASD has been hypothesized as a common underlying defect with many converging etiologies. For example, the impairment of gamma-aminobutyric acid (GABA) neurotransmission in autistic people is described by numerous reports [57,58,59]. These findings led to the choice of a mixture of pesticides (fipronil and pyriproxyfen), whose synergistic effect was correlated with the induction of GABA malfunction. Fipronil (FIP) is a compound from the phenylpyrazole class of insecticides that acts on the insect nervous system [60]. The main mechanism of action is the inhibition of GABA receptors and glutamate-gated chloride channels, and, depending of the dose, this can end in extreme neuronal excitation until the death of the organism [61,62]. Pyriproxyfen (PYR) is a compound that mimics the natural hormone for insect growth [63]. Both compounds, alone or in a mixture, are linked to developmental and histological abnormalities, behavioral disturbances, and elevated levels of oxidative stress [64,65,66,67,68,69,70,71,72,73].

Due to the increased role of the environment as a risk factor for ASD and the need for new therapeutic interventions for autistic people, the present study aimed to evaluate the effects of vit. of B_12_ and a mixture of pesticides administered individually or in combination in a zebrafish animal model, mimicking a real-life scenario. This objective was approached in the following steps: (1) evaluation of the effect of the compounds on locomotor activity, (2) characterization of sociability after exposure to the compounds, and (3) measurement of specific parameters of oxidative stress.

## 2. Materials and Methods

### 2.1. Animals

A total of 200 zebrafish juveniles (*Danio rerio*, WT AB, 2–3 months, ≈2 cm body length, 0.22 ± 0.05 g body weight, sex ratio 1:1) were obtained from an authorized local supplier and kept in the laboratory for three weeks as an acclimatization period. The water in the housing aquarium was replaced every 48 h with dechlorinated tap water, and the water in the experimental tanks was changed every single day. The water parameters were constantly measured and kept at normal values using filtration and an air pump (Table 1). The tanks were illuminated with LED bends (307.5 LUX) with a photoperiodic cycle of 14:10 h (light:dark). The fish were fed twice a day with TetraMin Tropical Flakes (5% of body weight in food per day, of which vit. B_12_ accounted for approximately 0.12 µg).

### 2.2. Chemicals

The pesticide mixture used in this study was purchased from a veterinary store in liquid form. The product purchased was mainly made up of the two pesticides (67.5 mg FIP and 67.5 mg PYR) but also contained other chemicals such as 0.3 mg of butylated hydroxyanisole, 60 mg of benzyl alcohol, and 0.15 mg of butylhydroxytoluene. The concentration of each pesticide compound was 600 μg L^−1^, which was achieved by dissolving a certain quantity of the previously prepared stock solution into the medium. Vit. B_12_ was bought from a local pharmacy as tablets of a recommended product for dietary supplementation, and the formulation contained microcrystalline cellulose, dicalcium phosphate, hypromellose, cellulose powder, magnesium stearate, and stearic acid in addition to the vitamin. The concentration of the vitamin used was 0.24 μg L^−1^. Each solution was prepared daily by grinding the tablets and dissolving the fragments in a 100 mL volumetric flask. Plastic vials with lids were filled with 40 mL of solution (prepared by dissolving a certain amount of the stock solution), and the fish were kept for 30 min according to Pena’s protocol [74]. The chemicals used in this protocol (the pesticide mixture and the vitamin tablets) were commercial compounds, since it is more common to use them that way than in the pure state of the active ingredients (to avoid conflicts of interest, the brands of the products will not be mentioned). The oxidative stress analysis was performed using the Superoxide Dismutase Determination Kit (SOD, 19160-1KT-F,); Glutathione Peroxidase Cellular Activity Assay Kit (GPx, CGP1-1KT,); and Total Protein Kit, Micro Lowry, Peterson’s Modification (TP0300-1KT) from Merck, Darmstadt, Germany. In addition, malondialdehyde (MDA) levels were assessed through the thiobarbituric acid-reactive substances assay according to the protocol of Balmus et al. [75].

### 2.3. Experimental Design

The fish were randomly transferred from the housing aquarium to the experimental tanks, with a capacity of 5 L each, and left to adjust to the new space. Four experimental groups were divided from each other as follows: untreated, 0.24 μg L^−1^ vit. B_12_, 600 μg L^−1^ FIP + PYR, and 0.24 μg L^−1^ vit. B_12_ + 600 μg L^−1^ FIP + PYR. Zebrafish juveniles were exposed to a single dose of the pesticide mixture, while vit. B_12_ was administered for a period of two weeks in order to simulate a real-life situation.

Following the accommodation period, the initial behavior of each fish was evaluated through the locomotor activity and social interaction tests.

A newly prepared pesticide solution was dissolved directly into the experimental tanks for the mixture and mixture + vit. groups. As regards vitamin administration, this was performed by placing each fish in a plastic vial filled with vit. B_12_ solution and leaving it in for 30 min. The control and mixture groups simulated exposure to the vitamin. Each fish was tested in the T-maze to collect data on locomotor activity and sociability as in the initial assessment of behavior. These tests were repeated daily for two weeks. Data were acquired every day using a camera situated above the maze, connected to a computer with EthoVision XT 11.5 software (Noldus Information Technology, Wageningen, Holland), with which all the behavioral parameters were calculated. In the end, fish were killed by immersion in ice-cold water for a minimum of 5 min after opercular motion had stopped and then stored in the freezer at −80 °C for oxidative stress analysis. The study included two more replicates. A schematic representation of the entire study is presented in Figure 1.

### 2.4. Behavioral Tests

#### 2.4.1. Locomotor Activity Test

The specific locomotor activity parameters of the animals were measured before and after the treatment using a T-maze adapted for this test. The experimental apparatus is made from transparent Plexiglass (40 × 30 × 10 cm, length × height × width) and divided into three arms: left, right, and center (Figure 2). The starting point was established at the end of the center arm. To investigate the impact of the compounds on animals, a series of parameters were chosen to describe the locomotor activity. The total distance swum was the first one chosen; it represents the total distance travelled by fish in the T-maze (cm) during a trial. Secondly, the average velocity parameter refers to how fast the fish is moving (cm s^−1^), while maximum acceleration is defined as the fish’s maximum speed of reaction (cm s^−2^). In addition, the time spent active or inactive describes the amount of time in which the fish was or was not moving (s). Each trial had a duration of 4 min, and an experimental session was carried out every day between 9 a.m. and 6 p.m.

#### 2.4.2. Social Interaction Test

The social interaction test aims to assess the tendency of an individual to choose and/or spend time next to its conspecifics. This test used the same experimental apparatus described in the previous subsection, but with several adaptations. A transparent wall was added in the left arm to divide the area into two zones: the social stimulus and the tested zone (Figure 3). In the social stimulus zone was placed a group of three animals (from the same group tested and with a different sex ratio every day), while the tested zone corresponded to the experimental fish. Each fish was allowed to swim freely in all arms, except for the social stimulus zone, which was separated. Social behavior was quantified by recording the time spent by fish next to the social stimulus zone in a session of 4 min. Beside this, the time spent in the center and right arms was measured.

### 2.5. Oxidative Stress Measurement

At the end of the chronic exposure, all the experimental fish were killed by immersion in cold water (under 5 °C) and kept in the freezer for the oxidative stress analysis. To measure the oxidative stress markers, each fish was defrosted and homogenized in ice-cold saline (0.90% NaCl). Afterward, all the samples were centrifuged at 5500 rpm for 15 min following Jin’s protocol [76]. The obtained supernatant was used to determine the SOD and GPx activity, MDA level, and protein concentrations. The enzyme activity was determined according to the suggested protocols from the kit packages and quantified with a Specord 210 Plus spectrophotometer from Analytik Jena, Jena, Germany, at the following specific wavelengths: 450 nm for SOD, 340 nm for GPx, and 532 nm for MDA. Protein was measured using the Bradford method and determined at 595 nm [77]. Each sample was tested in triplicate (Appendix A), and values were presented as the average ± SEM.

### 2.6. Statistical Analysis

OriginPro v.9.8 software (OriginLab Corporation, Northampton, MA, USA, 2021) was used to realize the statistical analysis for both experimental tests. The first step in analyzing the data consisted of verifying of the normality of the data distribution through the Shapiro–Wilk test. When normality was confirmed, Tukey’s post hoc test was applied to demonstrate the significant differences between group parameters before and after the treatment period. The α value was established at 0.05 to indicate the mean differences between the group, with the data being expressed as the average ± standard error of the mean (S.E.M.). The graphic presentation of the results for the locomotor activity and social interaction tests was generated using OriginPro software and Microsoft Package Excel files (Microsoft Office Professional Plus 2021) for oxidative stress results.

### 2.7. Ethical Note

The guidelines for the accommodation and care of animals used for experimental and other scientific purposes, Directive 2010/63/EU of the European Parliament, and the Council of 22 September 2010 on the protection of animals used for scientific purposes were strictly followed and maintained [78,79]. Additionally, this experiment was approved by the Ethical Commission of the Faculty of Veterinary Medicine, University of Agricultural Sciences and Veterinary Medicine Iasi, with registration number 750/04.07.2019.

## 3. Results

### 3.1. Short-Term Changes in Locomotor Activity Due to the Presence of Vitamin B_12_

The total distance swum by the control group did not reveal any changes during the experimental period (initial behavior: 886.3 ± 206.1 cm vs. the average of the study period: 886.6 ± 181.3 cm, *p* > 0.05, Tukey, ANOVA). Compared to the control group’s activity, the total distance for the 0.24 μg L^−1^ vit. B_12_ group registered increased values, with maximum peaks on D_5 (1518.3 ± 238.1 cm, *p* = 0.03, Tukey, ANOVA), D_7 (1333.4 ± 157.2 cm, *p* = 0.04, Tukey, ANOVA), and D_8 (1485.9 ± 152.8 cm, *p* = 0.04, Tukey, ANOVA) in contrast to the initial average: 755.1 ± 74.5 cm (Figure 4). A single exposure to the 600 μg L^−1^ FIP + PYR mixture triggered a decrease in the distance travelled in the first day (735.8 ± 166.4 cm, *p* > 0.05, Tukey, ANOVA) in comparison to the initial behavior (1090.1 ± 149.9 cm), but in the following days, the values of this parameter exhibited an upward trend—specifically, on D_2 (1638.5 ± 421.8 cm), D_5 (1592.6 ± 404.8 cm), and D_8 (1666.4 ± 294.5 cm)—without showing significant changes. Regarding the activity of the zebrafish exposed to the vitamin and pesticide mixture, the total distance value decreased on D_1 (707.2 ± 277.5 cm, *p* > 0.05, Tukey, ANOVA) vs. pretreatment (913.6 ± 53.8 cm), with ups and downs during the entire experimental period, ending at a greater value (1203.4 ± 323.8 cm) than was measured in the initial phase (Figure 4).

In regard to the parameter “swimming speed”, it presented similar trends, as can be seen in Figure 5. Swimming speed registered high levels for the 0.24 μg L^−1^ (initial behavior: 3.14 ± 0.31 cm s^−1^ vs. end of the study: 5.07 ± 1.07 cm s^−1^, *p* > 0.05, Tukey, ANOVA) and 600 μg L^−1^ FIP + PYR (initial behavior: 4.54 ± 0.62 cm s^−1^ vs. end of the study: 5.44 ± 0.81 cm s^−1^, *p* > 0.05, Tukey, ANOVA) groups compared to the control (initial behavior: 4.09 ± 0.70 cm s^−1^ vs. end of the study: 4.50 ± 0.34 cm s^−1^, *p* > 0.05, Tukey, ANOVA) and the remaining group (initial behavior: 3.78 ± 0.17 cm s^−1^ vs. end of the study: 5.01 ± 1.35 cm s^−1^ (*p* > 0.05, Tukey, ANOVA).

Another quantified parameter for the locomotor activity characterization was “maximum acceleration”. As can be seen in Figure 6, this parameter did not show important modifications in any group during the experimental period. In addition, the single exposure to the pesticide mixture triggered a slight increase in maximum acceleration on D_5: 235.8 ± 12.4 cm s^−2^ (*p* > 0.05, Tukey, ANOVA) compared to the pretreatment data: 211.9 ± 2.35 cm s^−2^. On the other hand, when the vitamin was also present, the parameter of maximum acceleration was reduced on D_1 (192.6 ± 19.16 cm s^−2^, *p* > 0.05, Tukey, ANOVA) and D_13 (191.1 ± 15.6 cm s^−2^, *p* > 0.05, Tukey, ANOVA) versus the pretreatment value: 223.2 ± 11.06 cm s^−2^.

The “active swimming” parameter presented several changes in the activity of the fish. Although most zebrafish individuals did not exhibit important modifications in this parameter’s values, a fluctuating trend was observed during the experimental period. The vit. B_12_ group was the only one with constant activity. Even though the 0.24 μg L^−1^ group showed constant activity, there was a decrease in it on D_2 (205.4 ± 8.54 s, *p* > 0.05, Tukey, ANOVA) in comparison to the pretreatment value: 217.6 ± 3.14 s (Figure 7). After the first week of exposure to the pesticide alone, fish were able to regain values close to the initial behavior (223.3 ± 7.43 s). Thus, the lowest values of this parameter were recorded on the treatment day (176.9 ± 41.1 s), D_2 (191.5 ± 31.7 s), and D_5 (199 ± 37.9 s). When fish were exposed to the vitamin and the pesticides, the active swimming parameter revealed unstable behavior. The highest value was recorded on D_12 (229.4 ± 3.91 s, *p* > 0.05, Tukey, ANOVA), and the lowest on D_1 (161.2 ± 32.7 s, *p* > 0.05, Tukey, ANOVA), compared to pretreatment (215.3 ± 10.2 s).

The ”inactive status” parameter revealed short-term changes in the experimental groups during the tested period. In the first week of exposure, as can be seen from Figure 8, the group exposed to 0.24 μg L^−1^ vit. B_12_ + 600 μg L^−1^ FIP + PIR presented a more pronounced time spent in inactivity compared to the initial behavior (average of the week: 62.36 ± 35.10 s vs. initial behavior: 24.64 ± 10.29 s). When the compounds were given separately, this trend changed. For instance, the vitamin group exhibited the lowest values for this parameter, in contrast to the pesticide group, which had several elevated peaks recorded for the treatment day (63.01 ± 41.1 s), D_2 (48.44 ± 31.7 s), D_5 (40.9 ± 37.3 s), and D_7 (41.3 ± 41.1 s). The control group did not show any modifications to this parameter (*p* > 0.05, Tukey, ANOVA).

### 3.2. Impact of the Presence of Vitamin B_12_ on Zebrafish Social Behavior

The sociability of zebrafish was evaluated through the social interaction test by measuring the time spent next to the stimulus zone. In Figure 9, the variations of this behavior during the experimental period are exhibited.

The control group revealed a normal and natural behavior that is specific to this organism, with the most time spent next to the stimulus zone (124.5 ± 31.8 s), followed by the center arm (88.5 ± 27.3 s) and then the right arm (26.9 ± 15.7 s), with no significant difference between the experimental period and pretreatment (left: 135.04 ± 35.3 s vs. center: 92.4 ± 30.5 s, vs. right: 12.6 ± 8.84 s, *p* > 0.05, ANOVA) (Figure 10).

Regarding the 0.24 μg L^−1^ B_12_ group, the time spent in the maze arms reflected elevated values for the left and center arms (average of tested period: 111.3 ± 22.6 s vs. 106.1 ± 20.2 s) and a reduced value for the right arm (22.6 ± 9.71 s) compared to initial behavior (left: 157.9 ± 16.1 s vs. center arm: 41.5 ± 8.5 s vs. right arm: 34.6 ± 14.6 s) (Figure 11).

Exposure to a single dose of 600 μg L^−1^ FIP + PIR did not trigger long-term effects on zebrafish sociability; however, this combination was able of disturbing this behavior after the first days of treatment. The time spent in the left arm decreased on D_1 (70.4 ± 43.1 s) and D_3 (96.7 ± 27.7 s) compared to the pretreatment period (124.1 ± 29.7 s). Thus, after these days, the fish started to regain their normal behavior, as can be observed in Figure 12. Additionally, the time spent in the right arm registered the lowest values (pretreatment: 20.2 ± 7.7 s vs. D_14: 14.1 ± 7.7 s), in contrast to the left arm (pretreatment: 95.7 ± 27.9 s vs. D_14: 75.1 ± 26.1 s) and center arm (pretreatment: 124.1 ± 29.7 s vs. D_14: 150.2 ± 33.1 s).

The group treated with 0.24 μg L^−1^ vit. B_12_ and 600 μg L^−1^ FIP + PIR spent less time in the left arm on the treatment day (96.02 ± 42.1 s), D_1 (70.4 ± 43.1 s), and D_2 (108.1 ± 35.7 s) than during the pretreatment period (120.4 ± 23.2 s). Starting on D_3, the time spent in the stimulus zone gradually increased, registering an average of 128.6 ± 30.8 s, almost matching that from the initial period (Figure 13). Regarding the other areas of the maze, the time spent in the right and center arms by the fish presented ups and downs with no significant activity (*p* > 0.05, Tukey, ANOVA) (Figure 13).

### 3.3. Antioxidant Boost after Vitamin B_12_ Supplementation

The biochemical activity is represented in Figure 14. A single exposure to the pesticide mixture did not trigger significant changes in SOD activity; neither did exposure to the pesticides and vit. B_12_ (*p* > 0.05 Tukey, ANOVA). The same trend was observed for GPx activity (*p* > 0.05 Tukey, ANOVA). In contrast, 14 days of treatment with 0.24 μg L^−1^ vit. B_12_ resulted in increased activities for SOD (*p* = 0.03 Tukey, ANOVA) and GPx (*p* = 0.02 Tukey, ANOVA). Regarding the lipid peroxidation process, the MDA marker did not show important variations among the experimental groups except for the last group (*p* = 0.03 Tukey, ANOVA), exposed to the pesticide mixture and the vitamin.

## 4. Discussion

The purpose of this study was to determine whether vit. B_12_ can act as a therapeutic tool in an ASD animal model developed through environmental risk factor exposure. Modeling traits comparable to those reported in autistic people in another organism implies multiple phases. The first step in the construction of an animal model is the discovery of a suitable inductor, followed by the validation of the existence of ASD and the evaluation of an adequate response after trying a recognized therapeutic method. The effectiveness of vit. B_12_ in treating symptoms induced by exposure to FIP and PYR (already established to have the potential to produce ASD-like impairments) was assessed in this trial. To date, no animal studies have been performed to evaluate the effects of vit. B_12_ on locomotor activity, on social features, or in association with ASD. In this study, the first of its kind, the findings reveal that vit. B_12_ can assist in restoring normal behavior in zebrafish after exposure to a mixture of pesticides by interacting with enzymes of the antioxidant system. As shown in the study results section, supplementing the zebrafish diet with vit. B_12_ led to improved levels of movement parameters describing fish locomotor activity.

Hyperactivity was one of the main findings recorded for the pesticide group, which presented high values for distance, velocity, and maximum acceleration parameters in comparison to the other experimental groups. Even when no significant, differences could be seen between the pretreatment and treatment phases. Although there was only a single high exposure to the mixture of pesticides, the impact could be seen over the whole study period, particularly during the first week. This may be explained by the time required for pesticides to be metabolized in fish. Furthermore, excessive movement can also be a response to the interaction of fish with these two pesticides, which causes the appearance of stress and triggers an increase in cortisol secretion but also in ROS [80,81]. This observation has been made in many research studies after FIP and PYR exposure due to the pesticides’ transformation into more toxic and lasting compounds. For example, the administration of a single oral dose of PYR (2 and 1000 mg kg^−1^) to rats showed that this pesticide had not only a higher excretion rate but also a reduced depuration period when the presence of tissue residues was determined after 7 days [82]. A more recent work, in which zebrafish were exposed to 10 and 100 μg L^−1^ PYR for 30 days, demonstrated the ability of the pesticide to accumulate after only one day of exposure, while the depuration period of 14 days indicated different half-life values: 2.3 days and 92.5% on day 7 for the lowest dose compared to 1.2 days and 94.6% on day 7 for the highest. In addition, compared with the other metabolites, both doses of PYR inhibited the activity of CAT and SOD in the liver [83]. Similar results as well as elevated levels of lipid peroxidation in the brain and kidney were obtained after oral administration of 2.5, 5, and 10 mg kg^−1^ body weight FIP to mice for 28 days [67]. The same observations were made in zebrafish after 96 h of exposure to 0.5, 1, and 2 mg L^−1^ FIP [82]. Compared with the previous study, in the present work, no significant effects were recorded on the activity of SOD, GPx, or MDA after pesticide exposure. This may be explained by variations in dose, exposure time, and developmental stage that may trigger different consequences on zebrafish. Moreover, when this pesticide mixture was used in a 14-day treatment for zebrafish, it resulted in increased activities for SOD, GPx, and MDA [83]. Consequently, it can be concluded that there is a link between the transformation of pesticides and the occurrence of oxidative stress, especially through the overproduction of free radicals, which can overwhelm the antioxidant system.

Although an effect of vit. B12 on the social behavior of zebrafish was not evidenced, oscillations between the left and central arms were observed after two weeks of treatment. On the other hand, the group exposed to 0.24 μg L^−1^ vit. B_12_ and 600 μg L^−1^ FIP + PIR showed a clear preference for the region with social cues, suggesting that vit. B_12_ may participate in certain fish metabolic processes, allowing them to behave as prior to treatment. This observation could lead to the conclusion that the vitamin does not directly participate or act on social behavior, but its involvement in other metabolic processes could impact the behavior of individuals. For example, compared to the results of the previous group, a single dose of the pesticide mixture was capable to induce short- and medium-term changes in fish sociability in the first week of exposure. This behavioral alteration did not last until the end of experimental period, indicating that pesticides and their effects began to fade via the excretion process as soon as fish were transferred to system water. Even if the differences were not validated by a specific degree of significance, the graphical representations indicate certain trends for the behavioral parameters studied, which must be further investigated.

The potential antioxidant effect of vit. B_12_ has been mentioned several times, but it remains unclear, though its involvement as a broad-spectrum micronutrient is known [84]. A recently published study discovered associations between subclinical vit. B_12_ deficiency and serum metabolic markers linked to neuronal and mitochondrial function, along with increased oxidative stress [85]. Moreover, vit. B_12_ interacts with superoxide, a product of aerobic metabolism, at rates comparable to SOD, which highlights the mimetic behavior of the vitamin towards the enzyme. [86,87]. This may indicate a possible mechanism through which the vitamin protects against chronic inflammation and controls redox homeostasis. In this study, exposure to 0.24 μg L^−1^ vit. B_12_ for 14 days resulted in elevated activities for SOD and GPx, whereas, when the vitamin was given after the pesticide mixture, the activity was lower than for control group. Similar to these results, a 7-day treatment with 0.63 g kg^−1^ vit. B_12_ reduced the levels of hepatic enzymes (aminotransferase and aspartate aminotransferase), increased antioxidant activities, and diminished inflammatory cell infiltration and necrosis processes in a male rat study of acetaminophen hepatotoxicity [88].

A possible explanation for this finding could be the ability of the vitamin administered in a non-stressful environment to promote and enhance the activity of the enzymes of the antioxidant system. On the other hand, in a disturbed environment (exposure to a mixture of pesticides) the vitamin participated together with the antioxidant system in counteracting the effects induced by FIP and PYR even with a single exposure. At the same time, the rate of accumulation in the zebrafish body and the products resulting from the metabolism of FIP and PYR may cause variations in the response to contact with the compounds. The ability of FIP to accumulate and the rate of metabolite formation in rainbow trout (*Oncorhynchus mykiss*) was measured after 32 days of exposure followed by 96 days of depuration. That study concluded that FIP is rapidly converted to its metabolite, fipronil sulfone, known to have a longer half-life [89]. For instance, the elimination half-life of FIP was 8.5 h, compared to 208 h for fipronil sulfone, after administration of 4 mg kg^−1^ FIP in rats [62,90]. Additionally, the persistence of fipronil sulfone was confirmed after a 7-day depuration process in European sea bass (*Dicentrarchus labrax*) juveniles previously fed with 10 mg kg^−1^ FIP for 14 days [91].

Another parameter measured to identify the existence of oxidative stress was MDA, a popular marker for lipid peroxidation. Exposure to a single dose of pesticides followed by 14 days of vit. B_12_ led to a decrease in the MDA level compared to the other groups. This result indicates a possible intervention of the vitamin to regulate the balance between ROS products and antioxidants, which is also supported by the decreased activities of SOD and GPx. Dietary supplementation with vit. B_12_ had a positive influence on the methylation process, GSH activity, and oxidative stress in autistic children after an 8-week course of therapy [43]. Similar results were previously reported by Bertoglio et al. following a 3-month course of therapy with 0.06 mg kg^−1^ vit. B_12_ [42]. A substantially decreased level of vit. B_12_ was recently found by two studies that assessed the vitamin’s activity in children with ASD [39,92]. Perturbation of vitamin activity can also occur due to genetic mutations, as demonstrated in zebrafish by mutations in the abcd4 gene, whose deficiency caused anemia, or the mmachc gene, responsible for processing the vitamin and transporting it to cells [53,93]. Vitamin deficiency was also proven in a mouse model of ischemic stroke, where both females and males had impaired balance and coordination as well as elevated homocysteine levels compared to the 0.025 mg kg^−1^ group [94]. Neurologic alterations such as anxiety, deficits in learning and memory, and changes in brain mass were obtained in a knockout mouse in which the transcobalamin receptor (TCblR) gene (CD320) was ablated [95].

Finally, the interaction among vit. B_12_, antioxidants, and ROS should be further studied and developed using animal models in an environmental setting, especially due to the high incidence of side effects, but also in a neuropsychiatric context.

## 5. Conclusions

The usefulness of the vitamin will need to be examined by developing animal models with similar characteristics to ASD, considering the new research that has come to light indicating a lack of vitamin B_12_ in people with autism, whether it is genetic or environmental. Furthermore, the present findings revealed the vitamin’s participation in an environment of oxidative stress, along with its contribution to zebrafish behavior. This study requires further attention to find out the optimal route of exposure, dose, and treatment duration; all of these are considered limitations of the current study.

## Figures and Tables

**Figure 1 animals-14-00405-f001:**
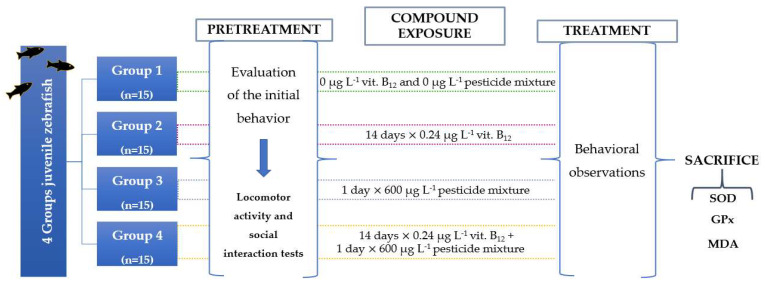
The experimental design of the study for evaluating the impact of vitamin B_12_ and pesticide mixture on zebrafish.

**Figure 2 animals-14-00405-f002:**
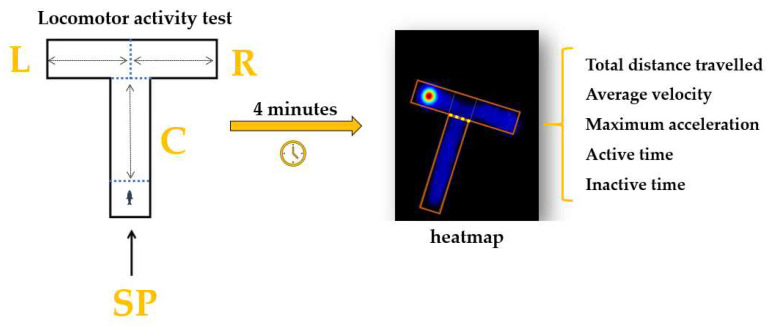
The T-maze adaptation for performing the locomotor activity test; C: center arm, L: left arm, R: right arm, and SP: start point.

**Figure 3 animals-14-00405-f003:**
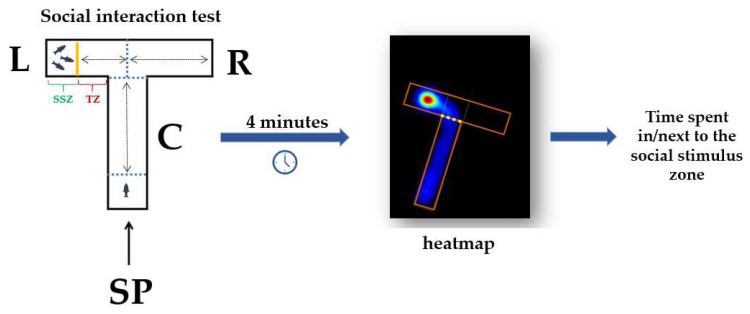
The T-maze adaptation for performing the social interaction test; C: center arm, L: left arm, R: right arm, SP: start point, SSZ: social stimulus zone, TZ: tested zone.

**Figure 4 animals-14-00405-f004:**
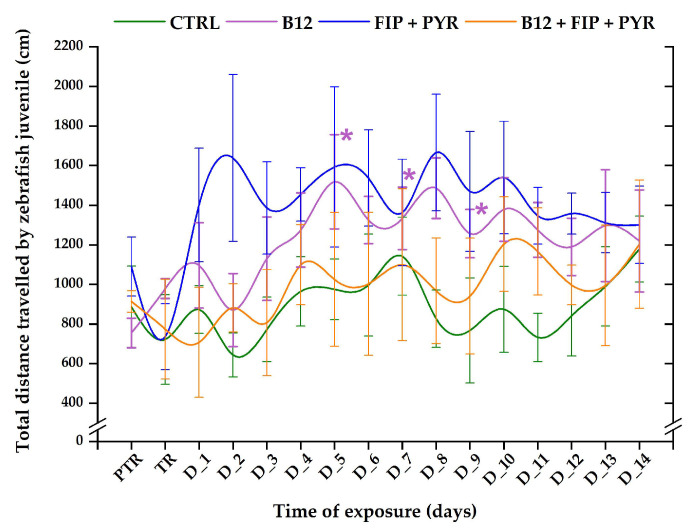
The total distance travelled by juveniles from experimental groups. D: day, PTR: pretreatment, TR: treatment, green: control, purple: 0.24 μg L^−1^ vit. B_12_, blue: 600 μg L^−1^ FIP + PIR, and orange: 0.24 μg L^−1^ vit. B_12_ + 600 μg L^−1^ FIP + PIR. The data are expressed as average ± SEM (n = 15); * *p* < 0.05 (ANOVA, Tukey) is significant compared to the results from the PTR stage.

**Figure 5 animals-14-00405-f005:**
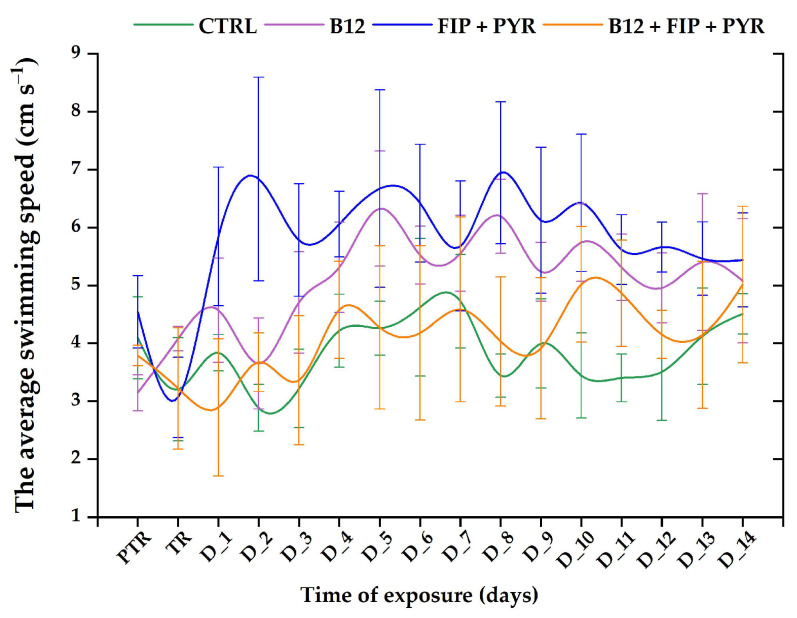
The average swimming speed of the experimental groups. D: day, PTR: pretreatment, TR: treatment, green: control, purple: 0.24 μg L^−1^ vit. B_12_, blue: 600 μg L^−1^ FIP + PIR, and orange: 0.24 μg L^−1^ vit. B_12_ + 600 μg L^−1^ FIP + PIR. The data are expressed as average ± SEM (n = 15); *p* < 0.05 (ANOVA, Tukey) is significant compared to the results from the PTR stage.

**Figure 6 animals-14-00405-f006:**
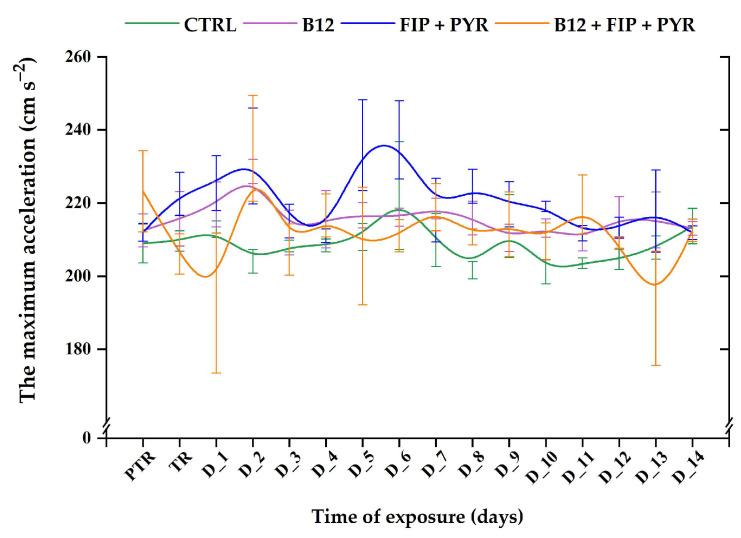
The maximum acceleration of the experimental groups. D: day, PTR: pretreatment, TR: treatment, green: control, purple: 0.24 μg L^−1^ vit. B_12_, blue: 600 μg L^−1^ FIP + PIR, and orange: 0.24 μg L^−1^ vit. B_12_ + 600 μg L^−1^ FIP + PIR. The data are expressed as average ± SEM (n = 15); *p* < 0.05 (ANOVA, Tukey) is significant compared to the results from the PTR stage.

**Figure 7 animals-14-00405-f007:**
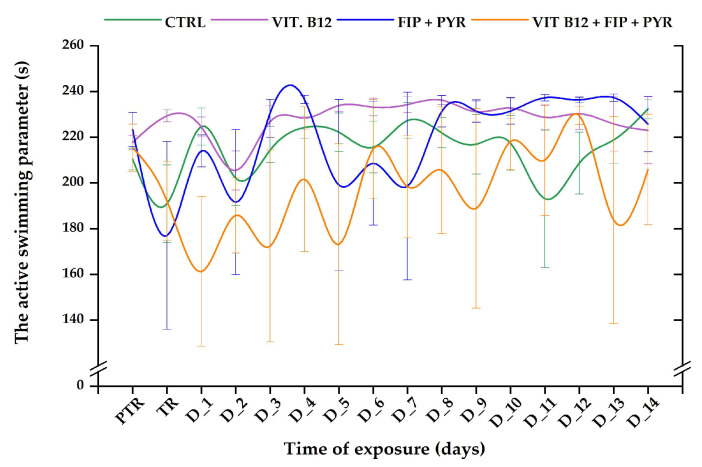
The active swimming parameter recorded for all the experimental groups. D: day, PTR: pretreatment, TR: treatment, green: control, purple: 0.24 μg L^−1^ vit. B_12_, blue: 600 μg L^−1^ FIP + PIR, and orange: 0.24 μg L^−1^ vit. B_12_ + 600 μg L^−1^ FIP + PIR. The data are expressed as average ± SEM (n = 15); *p* < 0.05 (ANOVA, Tukey) is significant compared to the results from the PTR stage.

**Figure 8 animals-14-00405-f008:**
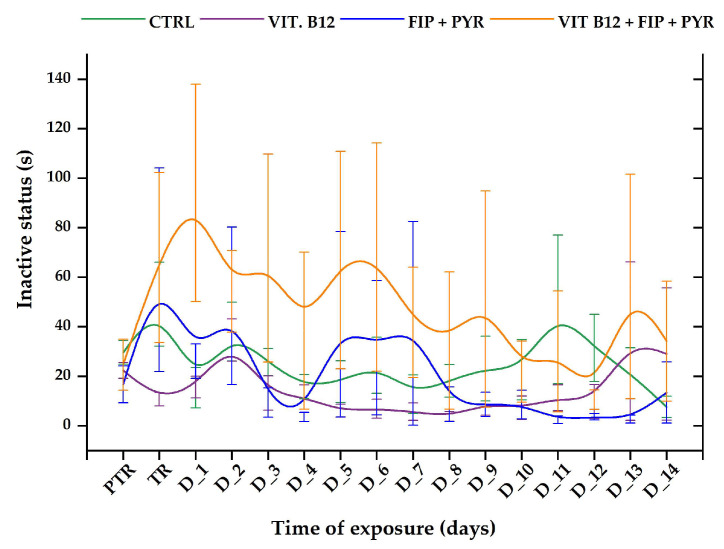
The inactive status recorded for all the experimental groups. D: day, PTR: pretreatment, TR: treatment; green: control, purple: 0.24 μg L^−1^ vit. B_12_, blue: 600 μg L^−1^ FIP + PIR, and orange: 0.24 μg L^−1^ vit. B_12_ + 600 μg L^−1^ FIP + PIR. The data are expressed as average ± SEM (n = 15); *p* < 0.05 (ANOVA, Tukey) is significant compared to the results from the PTR stage.

**Figure 9 animals-14-00405-f009:**
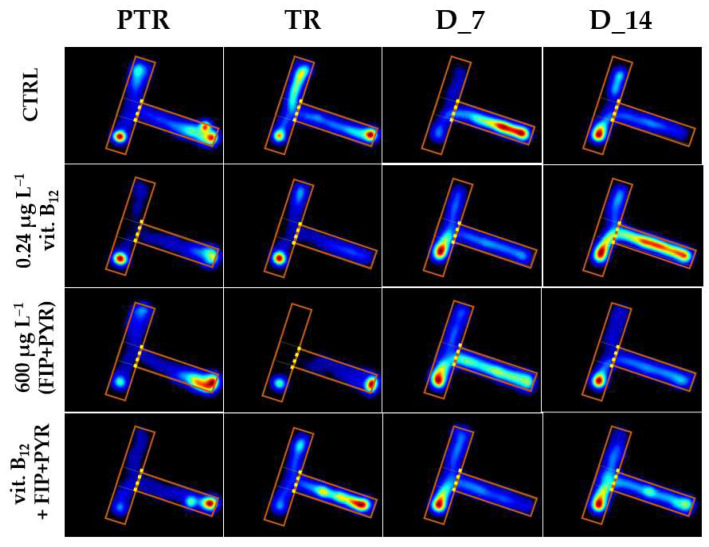
The time spent in the T-maze areas during the social interaction test, presented through heatmaps. D: day, PTR: pretreatment, TR: treatment.

**Figure 10 animals-14-00405-f010:**
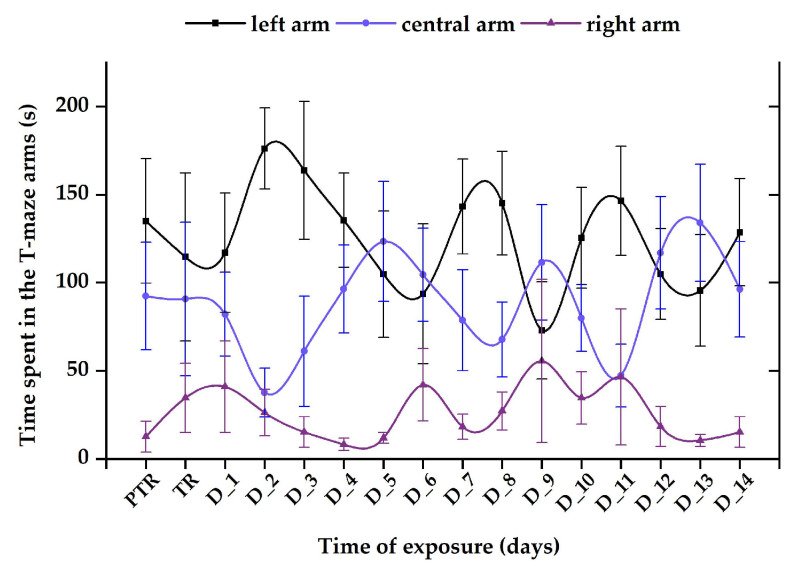
The time spent in the T-maze areas, recorded for the control group. D: day, PTR: pretreatment, TR: treatment. The data are expressed as average ± SEM (n = 15); *p* < 0.05 (ANOVA, Tukey_ is significant compared to the results from the PTR stage.

**Figure 11 animals-14-00405-f011:**
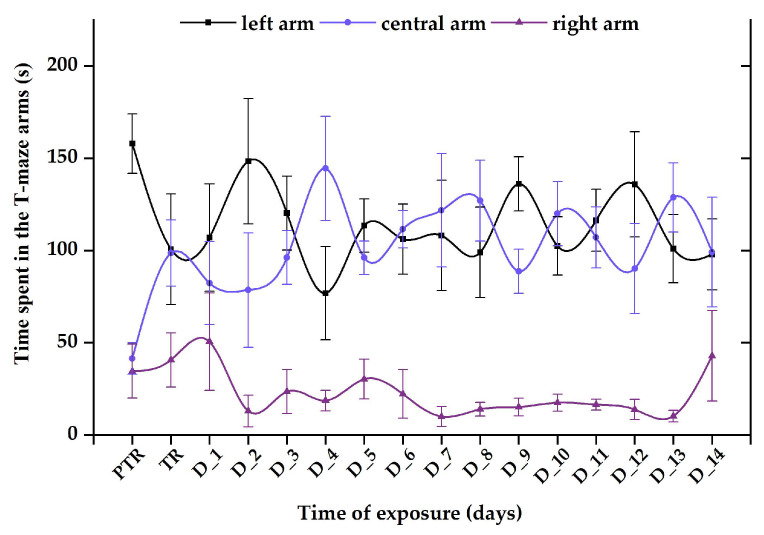
The time spent in the T-maze areas, recorded for the group exposed to 0.24 μg L^−1^ vit. B_12_. D: day, PTR: pretreatment, TR: treatment. The data are expressed as average ± SEM (n = 15); *p* < 0.05 (ANOVA, Tukey) is significant compared to the results from the PTR stage.

**Figure 12 animals-14-00405-f012:**
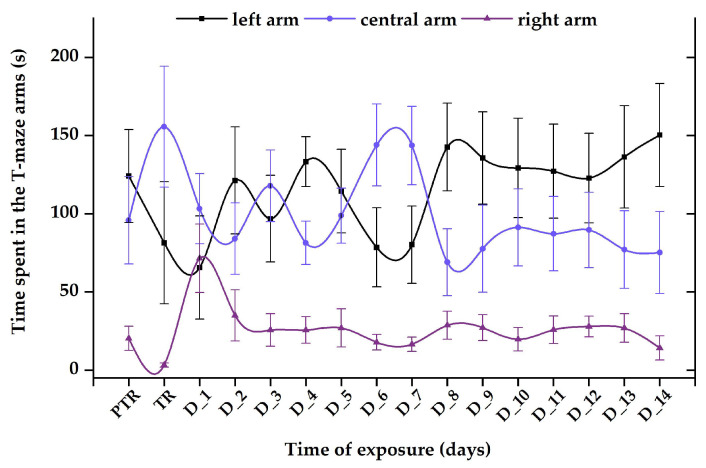
The time spent in the T-maze areas, recorded for the group exposed to 600 μg L^−1^ FIP + PIR mixture. D: day, PTR: pretreatment, TR: treatment. The data are expressed as average ± SEM (n = 15); *p* < 0.05 (ANOVA, Tukey) is significant compared to the results from the PTR stage.

**Figure 13 animals-14-00405-f013:**
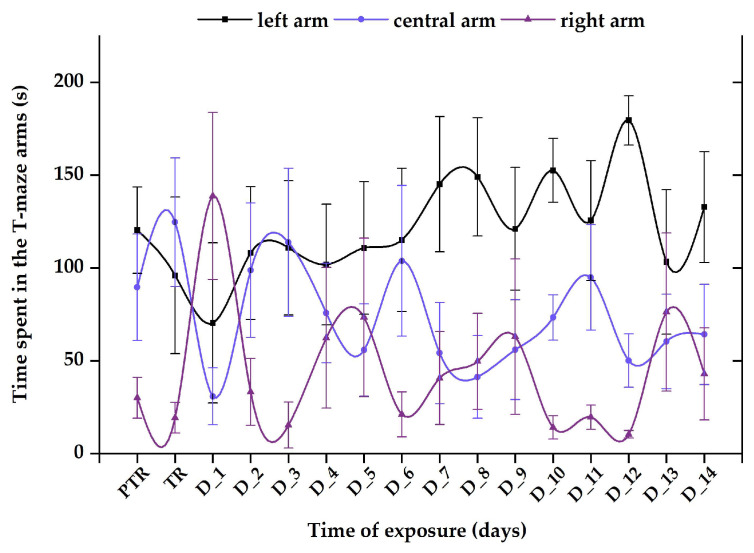
The time spent in the T-maze areas, recorded for the group exposed to 0.24 μg L^−1^ vit. B_12_ and 600 μg L^−1^ FIP + PIR mixture. D: day, PTR: pretreatment, TR: treatment. The data are expressed as average ± SEM (n = 15); *p* < 0.05 (ANOVA, Tukey) is significant compared to the results from the PTR stage.

**Figure 14 animals-14-00405-f014:**
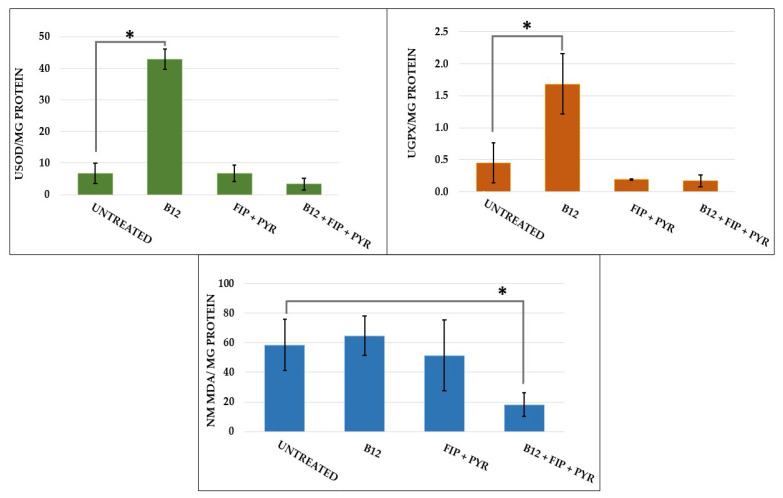
A graphical representation of the activity of superoxide dismutase (SOD), glutathione peroxidase (GPx), and malondialdehyde (MDA) for the experimental groups. The data are expressed as the average ± SEM; * *p* < 0.05 (Tukey) compared to the control.

**Table 1 animals-14-00405-t001:** Environmental conditions from the housing and experimental tanks observed during the experimental period.

Type of Tank	Temperature (°C)	pH	Conductivity (µS cm^−1^)	Salinity	Ammonia (mg L^−1^)	Dissolved Oxygen (%)
Housing tank	26 ± 0.5	7.6	551	0.26	0.05	90.8
Experimental tanks	25 ± 0.5	7.5	553	0.24	0.06	90.7

## Data Availability

Data supporting this study cannot be made available for ethical and commercial reasons.

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
