# Peer review of "Vitamin B12 Ameliorates Pesticide-Induced Sociability Impairment in Zebrafish (Danio rerio): A Prospective Controlled Intervention Study"

_animals, 2024, doi:10.3390/ani14030405_

Round 1

Reviewer 1 Report

Comments and Suggestions for Authors

I propose to complete the data related to the batch of fish used in the experiment with the assessment of the body mass of the individuals, the standard and total length.

Although the authors state that they use a system of pumping air into the pool, I believe that it is necessary to complete the list of monitored parameters with the concentration of oxygen in the water.

I propose to clarify the way of feeding fish and the concentration of vitamin B12 by relating the amount of food to the unit of volume or to the unit of body mass.

For the statistical processing of the data, I find it necessary to specify the number of measurements performed for the evaluation of oxidative stress markers in each type of sample.

Author Response

  • I propose to complete the data related to the batch of fish used in the experiment with the assessment of the body mass of the individuals, the standard and total length.

We added to the manuscript the required data as the reviewer indicated: ”≈2 cm body length, 0.22 ± 0.05 g body weight…”.

  • Although the authors state that they use a system of pumping air into the pool, I believe that it is necessary to complete the list of monitored parameters with the concentration of oxygen in the water.

We added a new parameter in Table 1 as the reviewer suggested regarding the concentration of oxygen in the water (dissolved oxygen: 90.8% in the housing tank and 90.7% for the experimental tanks).

  • I propose to clarify the way of feeding fish and the concentration of vitamin B12 by relating the amount of food to the unit of volume or to the unit of body mass.

Fish were fed daily with an estimated amount of food corresponding to 5% of the body mass of the fish per day. In consequence, we modified the text from the manuscript to: ”5% of body weight in food per day” instead of ”an average of 0.15 ± 0.05 g per portion per fish”.

  • For the statistical processing of the data, I find it necessary to specify the number of measurements performed for the evaluation of oxidative stress markers in each type of sample.

For the evaluation of oxidative stress markers, it was performed in triplicate for each sample, and the values shown in the manuscript represent the average of those three readings. Also, we added in the manuscript for a better understanding of the reader as: ”Each sample had triplicates, and values were presented as the average ± SEM”.

Reviewer 2 Report

Comments and Suggestions for Authors

The research was well conducted, reported and discussed. I recommend minor revision.

The introduction is rather detailed, but I did not see any hint on why a population might be in a B12 deficiency. A short note on this would be interesting.

Line 276-please correct the «0.24 μg L -1  group», detailing the vitamin supplementation here.

Line 400-the word ‘additionally’ is required here: «neither did the group treated “additionally” with vit. B 12 »

Author Response

  • The introduction is rather detailed, but I did not see any hint on why a population might be in a B12 deficiency. A short note on this would be interesting.

We took into consideration the reviewer’s suggestion, and we added to the introduction chapter the following: ”Despite the established recommended intake, which can vary across countries and continents, vit. B12 levels can record different values, that indicate a deficiency. For instance, children, adolescents, women of childbearing age, and the elderly are considered vulnerable for having a deficiency of vit. B12 according to Vargas-Uricoechea et al. [6]. With an under-recognized capacity to be considered a serious disorder, this deficiency can be heightened by an autoimmune condition, alcohol consumption, malabsorption, or a dietary insufficiency (e.g. vegetarians) [7].”.

  • Line 276-please correct the «0.24 μg L -1  group», detailing the vitamin supplementation here.

We completed it: ”the 0.24 μg L-1 vit. B12 group”.

  • Line 400-the word ‘additionally’ is required here: «neither did the group treated “additionally” with vit. B 12 »

This was done as indicated: ”neither did the group treated additionally with vit. B12”.

Reviewer 3 Report

Comments and Suggestions for Authors

The authors investigated vitamin B12 supplementation as a possible treatment for autistic spectrum disorder caused by pesticide poisoning. The zebrafish was chosen as the model organism. Based on the results of behavioural tests (locomotor activity and social interaction) and oxidative stress analysis (3 enzymatic assays), the authors concluded that vitamin B12 affects zebrafish behaviour and induces antioxidant enhancement.

Major concerns:

Materials and methods were poorly described. For example:

The authors stated that the 3- to 4-month-old zebrafish used in this study were still juveniles. This is a bit unusual as zebrafish should be reaching adulthood at this age. So my question is: what was the husbandry policy of the licensed breeder who supplied the fish? Why did these fish grow so slowly?

The names of all the chemicals used in this study and their manufacturers should be given, including those bought from a pharmacy (vitamin B12) and a veterinary shop (pesticides). Without this, the experiment cannot be repeated or fully evaluated. The reader is deprived of knowledge of the exact composition of the products used and cannot assess the possible influence of, for example, fillers on the experiment's outcome.

On line 171, the authors refer to Pena's 2017 protocol. As it stands, this description is misleading, as Pena et al. worked with young larvae. They did not use vitamin B12 and their PYR treatment was repeated daily. Therefore, the experiment had little in common with the presented work of Robea et al. When rewriting this part, the authors should specify how many fish were placed in the vial (in 40 ml). Even if it was one fish at a time, why was such a small volume used? Why was such a stressful condition chosen?

The experimental design (in 2.3) should be rewritten. Comparing the description with the graph in Figure 1, it seems that some text (lines 183-189) should be deleted. What was the rationale for choosing such small groups (n=15)? The fish in group 1 in Figure 1 cannot be called 'control' because they were not exposed to the same concentrations of additives listed in the VitB12 tables or to solvents used to prepare pesticide mixtures. Instead, this group should be called 'untreated fish'.

The authors should clarify what they mean when they say that 'the fish were randomly assigned'. Does this mean that the sex of the fish was not taken into account and that sex bias cannot be excluded? As the fish were fed twice daily, I assume that some tests were carried out on starved fish and some on freshly fed fish. I would like to know the order in which the fish were tested and whether the same schedule was used each day. Was the water changed, and the T-maze washed between fish? I would also like to know the dimensions of the T-maze.

From the description of the behavioural tests, I got the impression that the authors were comparing time spent / distance swum in zones of different sizes. The relatively small tested zone (TZ) should not be compared with the entire arms (especially since C seems larger than R). It would be good to mark the zones in Figure 2 to avoid confusion. Also, all heat map images should include an outline of the T-maze. Otherwise, the interpretation of this result is impossible.

There is no description of the three fish caught in the SSZ. Who were they? Were the same three fish used for the experiment (all 60 fish from groups 1-4), or did each group have its own set of 3 fish? How did these 3 SSZ fish react to the experimental fish? How was sex bias accounted for in this test?

Minor concerns:

Information on the volume of the tanks used in this study is missing.

It is unusual to have such precise information on the ammonia concentration in the water. How was it measured?

What solvent was used to prepare the stock solutions of the pesticides?

What about the two chemicals mentioned by the authors on line 172?

In its current form, I cannot recommend this manuscript for publication. However, once the description of the material and methods has been improved, the manuscript may be further revised.

Author Response

  • The authors stated that the 3- to 4-month-old zebrafish used in this study were still juveniles. This is a bit unusual as zebrafish should be reaching adulthood at this age. So my question is: what was the husbandry policy of the licensed breeder who supplied the fish? Why did these fish grow so slowly?

We thank to the reviewer for this observation. It was an error in typing; it should be 2-3 months instead of 3-4 months, but we changed it and added more details: ”2-3 months, ≈2 cm body length, 0.22 ± 0.05 g body weight, sex ratio 1:1)”.

  • The names of all the chemicals used in this study and their manufacturers should be given, including those bought from a pharmacy (vitamin B12) and a veterinary shop (pesticides). Without this, the experiment cannot be repeated or fully evaluated. The reader is deprived of knowledge of the exact composition of the products used and cannot assess the possible influence of, for example, fillers on the experiment's outcome.

We understand the reviewer’s opinion about mentioning the names of the chemicals used in this experiment, but we wish to keep them anonymous to avoid certain issues with the manufacturer. The chemicals used are the most common ones, and we added for each the rest of the components. The pesticide mixture comes from a veterinary product that contains 0.3 mg of butylated hydroxyanisol, 60 mg of benzyl alcohol, and 0.15 mg of butylhydroxytoluene, besides the active ingredients such as 67.5 mg fipronil and 67.5 mg pyriproxyfen. Concerning the vitamin tablets, microcrystalline cellulose, dicalcium phosphate, hipromellose, cellulose powder, magnesium stearate, and stearic acid were present along with 100 µg of vitamin B12. As for reproducing the present experiment, the components mentioned above are usually used in drug formulations and are known to be safe.

  • On line 171, the authors refer to Pena's 2017 protocol. As it stands, this description is misleading, as Pena et al. worked with young larvae. They did not use vitamin B12 and their PYR treatment was repeated daily. Therefore, the experiment had little in common with the presented work of Robea et al. When rewriting this part, the authors should specify how many fish were placed in the vial (in 40 ml). Even if it was one fish at a time, why was such a small volume used? Why was such a stressful condition chosen?

 Pena’s protocol was the only one found at the moment of designing the present experiment that was describing a method for vitamin exposure. Of course, we did not want to reproduce their protocol; we just took the exposure model and applied it to our juveniles. Moreover, each fish was exposed to the vitamin alone  for 30 minutes, as was highlighted in Pena’s protocol. Regarding the method of exposure, we chose this one to be sure that each fish is in direct contact with vitamin, knowing that fish can absorb compounds through the gills, mouth, and skin. Also, vials were selected based on previous tests to avoid creating a stressful environment for fish. We did not observe any sign of stress in the control group or in the exposed fish that could be related to the vial exposure.

  • The experimental design (in 2.3) should be rewritten. Comparing the description with the graph in Figure 1, it seems that some text (lines 183-189) should be deleted. What was the rationale for choosing such small groups (n=15)? The fish in group 1 in Figure 1 cannot be called 'control' because they were not exposed to the same concentrations of additives listed in the VitB12 tables or to solvents used to prepare pesticide mixtures. Instead, this group should be called 'untreated fish'.

We preferred to mention some of the details presented in the Figure 1 again to offer a better understanding to the readers, but we modified this section as the reviewer suggested: ”The fish were randomly transferred from the housing aquarium to the experimental tanks with a capacity of 5 L and left to get used to the new space. Four experimental groups were divided as: un-treated, 0.24 μg L-1 vit. B12, 600 μg L-1 FIP + PYR, and 0.24 μg L-1 vit. B12 + 600 μg L-1 FIP + PYR groups. Zebrafish juveniles were exposed to a single dose of the pesticide mixture, while vit. B12 was administered for a period of two weeks in order to simulate a real-life situation. Following the accommodation period, the initial behavior of each fish was evaluated through the locomotor activity and social interaction tests…”. Regarding the number of fish used in each experimental group, we intended to respect the housing number and the general regulations for the use of animals in research that indicate to use a minimum number of animals in experiments. Also, we changed control with untreated fish, as it was pointed out.

  • The authors should clarify what they mean when they say that 'the fish were randomly assigned'. Does this mean that the sex of the fish was not taken into account and that sex bias cannot be excluded? As the fish were fed twice daily, I assume that some tests were carried out on starved fish and some on freshly fed fish. I would like to know the order in which the fish were tested and whether the same schedule was used each day. Was the water changed, and the T-maze washed between fish? I would also like to know the dimensions of the T-maze.

The fish were randomly assigned, which means that we did not choose the fish depending on certain physical features (eg. the group was composed of 2-3 months old fish), but we respected the sex ratio (1:1, male:female) of the group components. Fish were fed every morning before testing and in the evening after finishing the tests. The testing schedule was the same for the entire period; the first testing was for social behavior and, secondly, the locomotor activity for untreated fish, vitamin group, pesticide group, and vitamin and pesticide group.  The water from the T-maze was changed after each group. Also, regarding the  dimensions of the T-maze, we completed the description of the method: The experimental apparatus is made from transparent Plexiglass (40 x 30 x 10 cm, length x height x width).”

  • From the description of the behavioural tests, I got the impression that the authors were comparing time spent / distance swum in zones of different sizes. The relatively small tested zone (TZ) should not be compared with the entire arms (especially since C seems larger than R). It would be good to mark the zones in Figure 2 to avoid confusion. Also, all heat map images should include an outline of the T-maze. Otherwise, the interpretation of this result is impossible.

As the reviewer indicated, we marked the zones in the figures to offer a better perspective of the experimental apparatus. We understand the reviewer’s concern regarding the data interpretation, but we evaluated all the parameters (for both tests) by comparing the obtained results during and after exposure with those recorded in the initial evaluation. Also, the outline of the T-maze was added to the figures.

  • There is no description of the three fish caught in the SSZ. Who were they? Were the same three fish used for the experiment (all 60 fish from groups 1-4), or did each group have its own set of 3 fish? How did these 3 SSZ fish react to the experimental fish? How was sex bias accounted for in this test?

For performing the social interaction test, the T-maze was adapted, and those three fish came from the same group studied (the method which was multiple times tested): ”In the social stimulus zone was placed a group of three animals (from the same group tested and with different sex ratios every day)…. Furthermore, the fish added in the social stimulus zone (SSZ) did not present significant activity during the tests, and the sex bias was controlled daily by varying the sex distribution as: two males and one female or two females and one male (we did not record changes depending on the group composition).

  • Information on the volume of the tanks used in this study is missing.

The volume of the tanks was 5 liters. As the reviewer indicated, we added this information to the manuscript: ”The fish were randomly transferred from the housing aquarium to the experimental tanks with a capacity of 5 L”.

  • It is unusual to have such precise information on the ammonia concentration in the water. How was it measured?

All the water parameters were measured using the Hanna multiparameter (HI-9828 Multi-Parameter Water Quality Portable Meter).

  • What solvent was used to prepare the stock solutions of the pesticides?

The pesticide solution was prepared using water as a solvent in order to avoid adding another chemical.

  • What about the two chemicals mentioned by the authors on line 172?

We were referring to the pesticide mixture and vitamin tablets, but our writing was a little bit confusing. Therefore, we changed the phrase from ”Both chemicals were commercial compounds since it is more common to use it like that than in the pure state of the active ingredient (to avoid conflict of interest, the brands of the products will not be mentioned).” to ”The chemicals used in this protocol (pesticide mixture and vitamin tablets) were commercial compounds since it is more common to use it like that than in the pure state of the active ingredient (to avoid conflict of interest the brands of the products will not be mentioned.”.

Reviewer 4 Report

Comments and Suggestions for Authors

I was honored to review the manuscript entitled “Vitamin B12 ameliorates pesticide-induced sociability impairment in zebrafish (Danio rerio): a prospective controlled intervention study” submitted to Animals.

I would like to thank authors for preparing this manuscript. The number of samples used in this study are fine. The manuscript’s structure is good and might be very interesting for the readers. However, there is some small points that should be clarified by authors.

There are some points to correct:

1.     Line 113 and 521: MMACHC should be written capital or in small character. In line 113 and 521 is in small but in introduction is written by capital character. Please correct.

2.     Citations are not according to journal’s format.

3.     Please clarify the oxidative stress measurement method as well as in which wavelength, the samples were measured?

4.     This study had two more replicates, please add some information about the result of the other groups. If the results have no significant difference, mention them in the result section.

5.      Did you use a pure vit B12 in this study? If this tablet countian other chemicals, please add them. Please clarify about the brand of the vit tablets and the dose of this tablet.

In conclusion, this manuscript can be published after minor revision.  

Author Response

I was honored to review the manuscript entitled “Vitamin B12 ameliorates pesticide-induced sociability impairment in zebrafish (Danio rerio): a prospective controlled intervention study” submitted to Animals.

I would like to thank authors for preparing this manuscript. The number of samples used in this study are fine. The manuscript’s structure is good and might be very interesting for the readers. However, there is some small points that should be clarified by authors.

  • Line 113 and 521: MMACHC should be written capital or in small character. In line 113 and 521 is in small but in introduction is written by capital character. Please correct.

We modified through the manuscript the indicated words: ”MMACHC” to ”mmachc”.

  • Citations are not according to journal’s format.

The citations were modified according to the journal’s format.

  • Please clarify the oxidative stress measurement method as well as in which wavelength, the samples were measured?

We used kit packages for superoxide dismutase, glutathione peroxidase, and malondialdehyde. The Bradford method was performed for protein measurement. We added: ”at the specific wavelengths (450 nm for SOD, 340 nm for GPx, 532 nm for MDA). Protein measurement was made using the Bradford method and determined at 595 nm.” as the reviewer kindly requested.

  • This study had two more replicates, please add some information about the result of the other groups. If the results have no significant difference, mention them in the result section.

Please check the supplementary material where it was added the other measurements for the oxidative stress markers. Also, this fact was mentioned in the manuscript in section 2.4: ”Each sample had triplicate (supplementary material)…”.

  • Did you use a pure vit B12 in this study? If this tablet countian other chemicals, please add them. Please clarify about the brand of the vit tablets and the dose of this tablet.

Our intention was to use the available therapeutic options found on the market. It is well-known that almost everyone supplements their diet with vitamins. So, we used tablets with vitamin B12 that were bought from a local pharmacy and, of course, free to purchase by everyone. Regarding the brand of the vitamin supplement, we wish to keep it anonymous to avoid further issues with the manufacturer of the tablets. Although the brand of the tablets can’t be mentioned, we added the rest of the components of the tablet besides vitamin such as: ”microcrystalline cellulose, dicalcium phosphate, hipromellose, cellulose powder, magnesium stearate, and stearic acid.”